# Ionospheric S4 Scintillations from GNSS Radio Occultation (RO) at Slant Path

**Dong L. Wu**

NASA Goddard Space Flight Center, 8800 Greenbelt Rd, Greenbelt, MD 20771, USA; dong.l.wu@nasa.gov

**Abstract:** Ionospheric scintillation can significantly degrade the performance and the usability of space-based communication and navigation signals. Characterization and prediction of ionospheric scintillation can be made from the Global Navigation Satellite System (GNSS) radio occultation (RO) technique using the measurement from a deep slant path where the RO tangent height ($h_t$) is far below the ionospheric sources. In this study, the L–band S4 from the RO measurements at $h_t$ = 30 km is used to infer the amplitude scintillation on the ground. The analysis of global RO data at $h_t$ = 30 km shows that sporadic–E (Es), equatorial plasma bubbles (EPBs), and equatorial spread–F (ESF) produce most of the significant S4 enhancements, although the polar S4 is generally weak. The enhanced S4 is a strong function of local time and magnetic dip angle. The Es–induced daytime S4 tends to have a negative correlation with the solar cycle at low latitudes but a positive correlation at high latitudes. The nighttime S4 is dominated by a strong semiannual variation at low latitudes.

**Keywords:** GPS radio occultation; ionospheric scintillation; electron density; plasma bubbles; polar scintillation; solar cycle

## 1. Introduction

Irregular structures and variations in plasma density often cause scintillation of radio wave communication in a transionospheric link. Because changes of the refractive index are proportional to electron density gradient, as well as to the inverse of radio wavelength squared, ionospheric scintillations are strongly scale and frequency–dependent. Causes of ionospheric scintillations may be diffractive and/or refractive, depending on whether the real or imaginary part of refractive index dominates the process. In the case where the scale of inhomogeneity is comparable with radio wavelength, signal amplitude fluctuations are diffractive, or producing amplitude (S4) scintillation. In the case where there is a rapid temporal variation in the plasma refractivity, fluctuations in the phase measurement are refractive, inducing phase ($\sigma_\phi$) scintillation due to cycle slips or loss of phase lock [1]. In reality, strong amplitude fluctuations may result in elevated phase measurement errors, for which amplitude and phase scintillations occur simultaneously.

For GNSS (Global Navigation Satellite System) radio occultation (RO) applications, recent studies have been focusing on the S4 scintillations, using signal–to–noise (SNR) fluctuations induced by sporadic–E (Es) [2–4] and by equatorial plasma bubbles (EPBs) in the F–region [5–7]. While ground–based receivers can resolve variations of ionospheric disturbance structures with a high spatiotemporal resolution [8–10], the S4 data are mostly limited to observations over lands, leaving vast oceanic and polar regions uncovered. Thus, spaceborne in–situ [11–18] and remote sensing [5–7] techniques have been employed to obtain a global coverage as well as an altitude coverage in the ionosphere/thermosphere to better understand where the scintillations are originated and their underlying growth and decay processes.

Because ionospheric scintillations can significantly degrades the performance of transionospheric communication and navigation systems on the ground, a considerable amount of efforts have been

devoted to connect the L–band S4 scintillations observed from space to those experienced on the ground [17–22]. These studies show satellite in–situ measurements tend to yield a better correlation with the ground–based scintillations than the RO–based measurements. Several factors may cause differences between the S4 scintillation intensities observed from space and on the ground, including plasma density gradient, F–region background density [5], satellite measurement altitude [17], and amplitude overestimation [22].

This study presents a detailed analysis of S4 scintillation from the RO data at tangent heights ($h_t$) far below the ionospheric density disturbances. Here the RO tangent height is defined as the tangent point of straight–line height above the surface. It is argued that the S4 scintillations from the deep $h_t$ are more relevant to the scintillations experienced on the ground than those RO measurements from a higher $h_t$. The scintillation measurements from the deep $h_t$ share a similar viewing geometry as a ground–based receivers in terms of sensitivity to vertically–tilted density gradients and small–scale horizontal structures.

## 2. Data and Method

### 2.1. 50–Hz RO Data

In this study we analyze the atmPhs and conPhs data that contain 50–Hz signal–to–noise (SNR) and excess phase profiles. The excess phase reported in the atmPhs and conPhs files is the additional phase delay/advance in RO technique, which is typically due to ionospheric/atmospheric effects, after the contributions from GPS (Global Positioning System) transmitter and receiver satellite motions are removed. The correction still leaves an arbitrary constant in excess phase. As a result, the RO excess phase often references its profile to its top by setting the value at the top $h_t$ to zero. Different from atmPhs, the conPhs data apply navigation bits to ensure that the excess phases are connected. These files contain additional variables such as tangent height, longitude, latitude, and UTC. Table 1 lists the four RO data sets used in this study, which are obtained from the archive published at UCAR (Corporation for Atmospheric Research) CDAAC (COSMIC Data Analysis and Archive Center).

**Table 1.** Global Navigation Satellite System (GNSS) radio occultation (RO) data used in this study.

| LEO Satellites | Mission Lifetime | GPS RO Data | Sat Alt (km) | Sun–syn | RO Top Ht (km)[†] | Max No. ROs/Day | Lat Coverage |
|---|---|---|---|---|---|---|---|
| COSMIC1–1 | 2006–2018 | 2006–2018 | 525,810 | No | ~130 | 750 | global |
| COSMIC1–2 | 2006–2016 | 2006–2016 | 525,810 | No | ~130 | 680 | global |
| COSMIC1–3 | 2006–2010 | 2006–2010 | 525,725 | No | ~130 | 750 | global |
| COSMIC1–4 | 2006–2015 | 2006–2015 | 525,810 | No | ~130 | 750 | global |
| COSMIC1–5 | 2006–2017 | 2006–2017 | 525,810 | No | ~130 | 700 | global |
| COSMIC1–6 | 2006– | 2006– | 525,810 | No | ~130 | 670 | global |
| MetOp–A | 2006– | 2007– | 820 | Yes | ~85 | 730 | global |
| MetOp–B | 2012– | 2013– | 820 | Yes | ~85 | 710 | global |
| MetOp–C | 2018– | 2019– | 820 | Yes | ~85 | 670 | global |
| COSMIC2–1 | 2019– | 2019– | 545 | No | ~100 | 1100 | < 45° N/S |
| COSMIC2–2 | 2019– | 2019– | 715,545 | No | ~100 | 1110 | < 45° N/S |
| COSMIC2–3 | 2019– | 2019– | 715 | No | ~140 | 1050 | < 45° N/S |
| COSMIC2–4 | 2019– | 2019– | 715,535 | No | ~140 | 1100 | < 45° N/S |
| COSMIC2–5 | 2019– | 2019– | 715 | No | ~100 | 1060 | < 45° N/S |
| COSMIC2–6 | 2019– | 2019– | 715 | No | ~100 | 1080 | < 45° N/S |
| C/NOFS | 2008–2015 | 2010–2015 | 640–380 | No | ~170 | 290 | < 25° N/S |

Note: [†]RO top $h_t$ is the approximately highest tangent height where the 50–Hz RO begins.

Launched in 2006, COSMIC–1 (Constellation Observing System for Meteorology Ionosphere and Climate–1) is a 6–satellite constellation mission with a diurnal pole–to–pole coverage. The number of COSMIC–1 RO profiles has significantly reduced since 2016, with only one satellite in operation. The 50–Hz RO sampling starts from $h_t$ ~130 km to a height below the surface. This top height allows



the 50–Hz profiles to detect most of the Es layers that occur in the 90–120 km altitudes. Each COSMIC–1 satellite has two receivers in the forward and backward flight direction, together producing as many as ~720 occultations/day by tracking GPS L1, L2, and L2C signals. It becomes the standard design nowadays for LEO (low–Earth orbit) GNSS receivers to double the coverage by taking both rising or setting ROs. Usually, the rate of these ROs is ~2 km/s, producing a 50 m vertical resolution in the 50–Hz data.

COMSIC–2, a follow–on mission to COSMIC–1, also with 6–satellite constellation, was launched in 2019 to cover latitudes between 45° S–45° N [23]. COSMIC–2 is still in its commissioning phase where satellite constellation, orbit altitude, and RO acquisition are subject to changes before the final configuration is completed. Like COSMIC–1, each satellite is equipped with two RO receivers, called Tri–band GNSS Receiver System (TGRS) [24], capable of tracking legacy and new GPS signals, such as L5, L2C, and L1C/A, GLONASS (GLObal NAvigation Satellite System L1C, L1P, L2C, and L2P), and Galileo (E1B/C and E5A) with much improved sensitivity. COSMIC–2 has produced as many as 1100 occultations/day/satellite from GPS and GLONASS [25] and demonstrated a total of 6200+ GNSS occultations/day. Although COSMIC–2 receivers record on–board S4 measurements with a specified threshold, in this study we chose to derive S4 from the 50–Hz data without imposing any threshold for counting elevated events.

Sun–synchronous polar orbiting MetOp satellites, flown successively since 2006, are European operational Earth observing systems for meteorology and climate studies. Like COSMIC RO observations, MetOp–GRAS (GNSS Receiver for Atmospheric Sounding) makes RO measurements in the forward and backward flight directions. As shown in Table 1, each MetOp satellite is capable of producing more than 700 occultations/day, with nearly uniform geographical distribution from pole to pole. All MetOp satellites had been station–kept at the similar sun–synchronous orbits with stable daily sampling for ROs, which provides a valuable dataset for long–term variability studies at two fixed local times sampled by MetOp. Starting in August 2018, the MetOp–A satellite orbit begin to drift slowly from its fixed equator–crossing time, preparing for end of lifetime (EOL) operation. The climatological records at these local times are being carried on by the MetOp–B and –C systems.

Communication/Navigation Outage Forecasting System (C/NOFS), launched in April 2008 and ended in November 2015, is a prototype of an operational system to forecast the ambient ionosphere and scintillations. The goal of the C/NOFS mission is to detect active scintillation and forecast areas of scintillation probability in the equatorial region. The C/NOFS satellite has a low (13°) inclination elliptical (401 × 867 km altitude) orbit initially with a single RO receiver onboard, but the average orbital altitude had decreased gradually during 2010–2015 from 640 to 380 km. The C/NOFS receiver was operated to produce an extensive height range (up to 170 km) with the 50–Hz occultation, which is helpful to further isolate scintillation sources between the F–region and the E–region. The mission has two phases, survey mode and forecast mode. During survey mode the C/NOFS sensors will collect as much data as possible to identify the key parameters for predicting scintillations. C/NOFS was operated in survey mode for the first few months of the mission, then transitioned to forecast mode for the remainder of the payload operations. The C/NOFS data used in this study are from 2010–2015.

## 2.2. S4 Calculation and Aggregation

In this study the S4 index is defined as 1–s scintillation from the 50–Hz RO SNR data, using the conventional definition

$$S4 = \sqrt{\frac{\langle I^2 \rangle - \langle I \rangle^2}{\langle I \rangle^2}} \tag{1}$$

where $I = (SNR/SNR_0)^2$ is the intensity of RO signal normalized by the mean $SNR$ at $h_t > 40$ km. The SNR from CDAAC is reported as a ratio in receiver voltage unit (V/V in 1 Hz) and the RO signal power is proportional to voltage amplitude squared. Since the RO signal at $h_t > 40$ km is nearly constant (namely, $SNR_0$) in the absence of scintillations, the normalization provides a fair characterization of

scintillations between strong and weak RO signals by dividing $SNR$ by $SNR_0$. To minimize impacts of noisy signals on S4 statistics, the RO profiles with a low $SNR_0$ ($SNR_0 < 100$ V/V–Hz, or 20 dB–Hz) are excluded. A 1–s running mean is employed to compute the time series of $\langle I \rangle$. The S4 derived from the running, unlike the boxcar–averaged values, allows to record the scintillation more accurately with respect to its occurrence tangent height.

The derived S4 profiles are further aggregated into 2–hourly local time, 2–km height, $4° \times 8°$ latitude–longitude bins for monthly climatology. The 2–km height bin size is roughly equivalent to the 1–s running mean as used above for the perturbation calculation, because most ROs have a rising/setting rate of 2 km/s. The monthly aggregation was carried out for S4 power, i.e., $\langle (S4) \rangle^2$, to preserve the scintillation power during statistical averaging. Finally, the monthly mean is reported as $\overline{S4} = \sqrt{\langle (S4) \rangle^2}$, which is the variable presented in the rest of the paper. No threshold is imposed for compiling the S4 climatology in this study. Unlike occurrence frequency statistics reported in other studies [7,15], the S4 climatology from normalized $I = (SNR/SNR_0)^2$ and threshold–less aggregation includes weak–but–frequent scintillation events as well.

### 2.3. RO Slant Views and Smearing Effects on Scintillations

The RO scintillations at a deep tangent height (e.g., far below F–region irregularities) can provide an indication for the scintillations experienced by the ground–based GNSS receivers. An earlier study illustrated the importance of RO ray path with respect to ionospheric disturbance structures [2]. As shown in Figure 1a, strong scintillations occur where the RO ray path is parallel to either tilted or stratified density perturbation structures. In the tilted case (Type A), such as those induced by F–region bubbles and spread–F, scintillations can occur in the RO profile at a tangent height below the F–region disturbances, as the occultation ray path becomes aligned with the structures. For example, the RO signals at $h_t = 30$ km can experience a strong scintillation if the F–region disturbances are tilted by ~18°. For the RO signals at $h_t = 330$ km, disturbances at 400 km need to have a tilt angle of 8° in order to cause a significant scintillation. However, layered structures (Type B) at 330–350 km that are stratiform or with a tilt angle <5° can induce strong scintillations on the RO signals at $h_t = 330$ km if the RO ray passes through these layers near the tangent point.

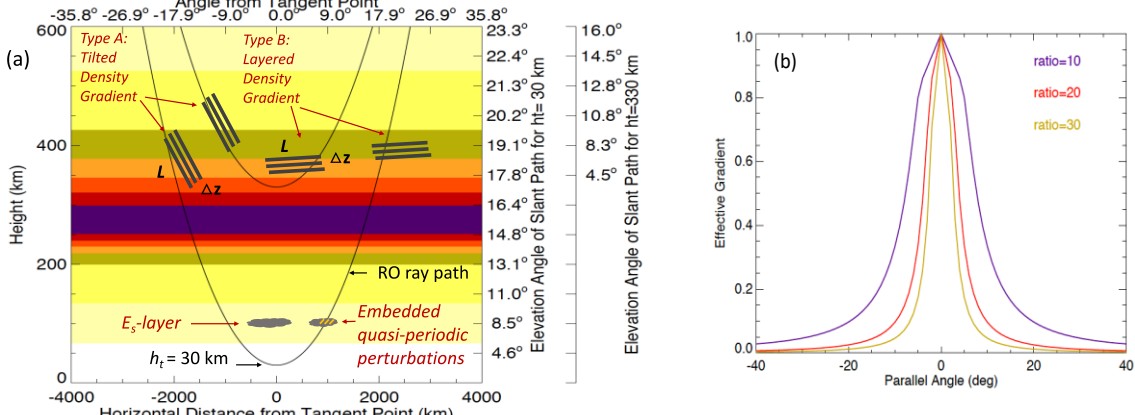

**Figure 1.** (**a**) Slant view geometry for the GNSS RO observations at $h_t = 30$ km and $h_t = 330$ km. The elevation angles for $h_t = 30$ and 330 km are shown in the right as a function of layer altitude. (**b**) Effective layer gradient from sinusoidal waves as a function of parallel angle and the ratio of smearing length ($L$) and layer thickness ($\Delta z$), i.e., $L/\Delta z$. Depending on the tilt angle of structured ionospheric disturbances, their impacts on the observed scintillation amplitude may be different. Scintillations from a deep slant view are more sensitive to the tilted than layered density gradients.

The S4 amplitude is sensitive to the angle at which the RO ray path intercepts plasma disturbance structures. Because scintillation intensity is proportional to the gradient of ionospheric density layers,

in Figure 1b we show the effect of the smearing by RO ray paths as a function of parallel angle and the ratio of smearing length and layer thickness. The smearing length (*L*) is the RO ray path length intersecting the disturbance structures. The layer thickness ($\Delta z$) is a characteristic scale of spatial separation of density disturbances. For illustration purposes, a periodical sinusoidal oscillation is used with $\Delta z$ as the wavelength. Depending on the angle of a RO transversal with respect to disturbance layers and the smearing ratio ($L/\Delta z$), the "effective" density gradient 'seen' by RO may be different. In reality, plasma density perturbations often do not have a layer–like structure, but the scale analysis with the smearing ratio ($L/\Delta z$) can be used as the first–order principle to evaluate RO responses to those complex fluctuation structures. A scale analysis was provided by Costa and Kelley [26], showing that S4 can vary with the size and structure of plasma irregularities.

　　This "effective" density gradient, or path–integrated structure, is the key concept that helps to understand the scintillation intensity observed in a GNSS RO profile. In the case that the RO ray is parallel to the layers (Type B), or zero parallel angle, the RO rays would experience a layer gradient similar to the original, where the disturbance layers have a maximum impact on the scintillation. Stratiform Es layers are a good example of Type B cases, of which the thickness is usually 1–2 km. However, quasi–periodic (QP) density fluctuations are often embedded in an Es layer that manifest themselves as field–aligned radar echoes [27]. In either case, if the RO ray intersects these embedded structures at a parallel angle, the "effective" density gradient can decrease sharply with the angle due to smearing by the RO path. As shown in Figure 1b, for a smearing ratio of 30, the "effective" density gradient can drop by an order of magnitude at the parallel angle of ~7°. A smearing ratio of 30 is not unreasonable for typical observed spread–F and bubble sizes [28–30] and a smearing length of 200–600 km by RO.

　　Figure 2 illustrates the scintillations observed by COSMIC–1 ROs as a function of straight–line height (SLH), or $h_t$, showing an example of S4 characteristics induced by different ionospheric/atmospheric density structures in January from 2006–2014. SLH is close to at $h_t > 30$ km, but deviates from $h_t$ below that due to atmospheric bending. Es layers are a dominant source of RO scintillations at $h_t = 80$–120 km. However, their impacts do not extend down to the lower tangent heights with the same intensity, due to the smearing–angle effect discussed in Figure 1. As shown in an earlier study by Wu et al. [2], however, the density fluctuations embedded in Es layers may have a tilted structure, which help to extend the Es–layer scintillations to a lower altitude. If all Es layers were perfectly stratified without embedded wave structures, their effect on RO scintillations would exhibit a sharp decrease as the tangent height moves below the layers, as seen in the case of the tropical tropopause layer (TTL) in Figure 2 (left panel). The TTL–induced scintillations have a very sharp drop below the layer, suggesting that TTLs are highly stratified and their interactions with RO propagation are confined in a narrow height range. In summary, the Es effects on L–band scintillations are generally weak on the ground, but not negligible due to embedded wave structures. These stratiform, sometimes mirror–like, high electron density layers allow occasional long–distance communications at 20–150 MHz on the ground, and loss of ground–satellite contacts at some elevation angles.

　　The RO S4 scintillation can sometimes be present throughout the entire height range. For example, in the region of 10° W–60° W longitude and 30° S–30° N latitude, as shown in the right panel of Figure 2, the S4 enhancement extends to a much lower height and likely to the ground level at local solar time (LST) between 20:00–24:00. This extensive S4 distribution is indicative of a broad scale of disturbances and structures such as F–region bubbles and spread–F. In the event of F–region bubbles, their irregular structures can intersect all RO ray paths with a significant 'effective' density gradient, causing scintillations in a wide range of tangent heights. For the same reason, the intersection with the bubbles also causes scintillations in the ground–based receivers with different elevation angles. As a result, the RO scintillations at a deep tangent height (e.g., 30 km) serves as a good proxy for the scintillations experienced by the ground–based GNSS receivers. The extensive S4 scintillations are found in all monthly climatology (Figure S1 – Supplementary), but the enhancement in the (10° W–60° W, 30° S–30° N) region is perhaps weaker in May–August from 2006–2014.

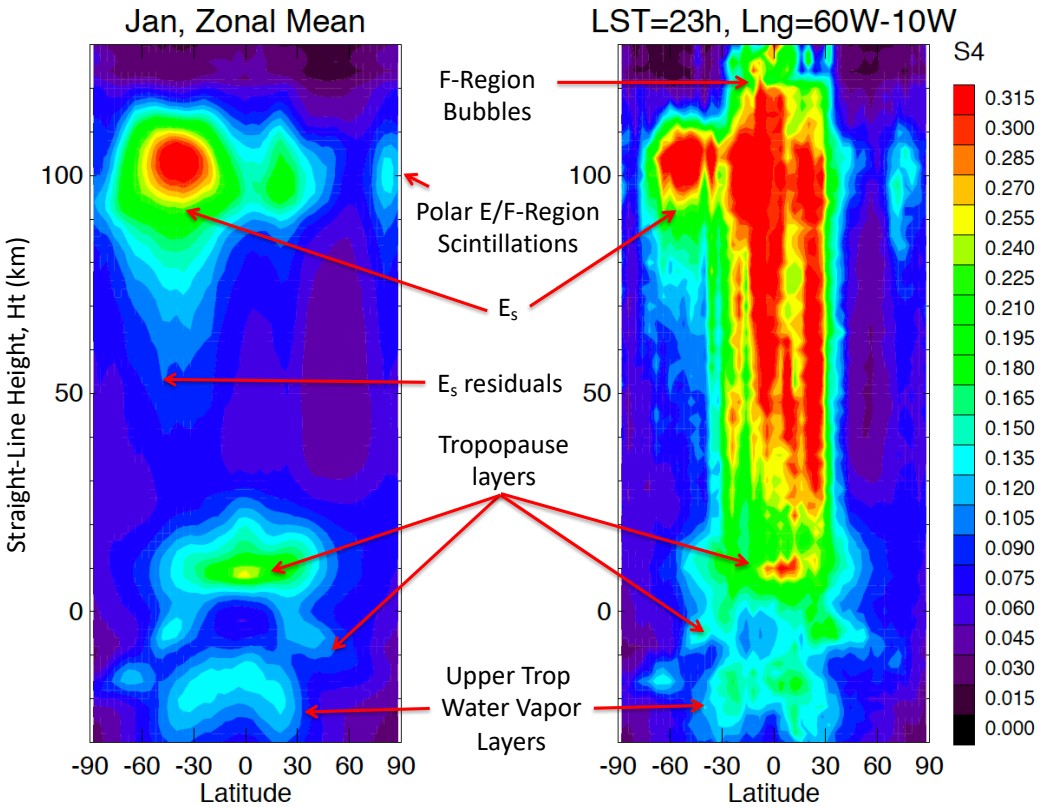

**Figure 2.** COSMIC–1 S4 profiles as a function straight–line height (SLH) for January of 2006–2014 (left: all– local solar time (LST), all–longitude zonal mean; and right: a regional mean from 20:00–24:00 LST and longitudes between 10° W–60° W).

The $h_t$ = 30–km is perhaps the lowest RO tangent height to measure ionosphere–induced scintillations before the RO signals are impacted by the scintillations from the neutral atmosphere. The sharp temperature gradient near the tropopause is responsible for the scintillations seen by RO in the tropics and the extratropics. Beneath the tropopause, the sharp increase in water vapor abundance in the middle and lower troposphere indicates the top of the hydrosphere, which magnifies RO scintillations related to atmospheric moisture layers. Therefore, in this study we use the 30 km scintillations as a proxy of the ground–based scintillations from the ionosphere.

## 3. Results

### 3.1. Sporadic–E (Es)

Although the S4 intensity from Es and auroral E tends to decrease with $h_t$, their effects on ground–based receivers may still be significant depending on the structure and variability of these E–region irregularities. Because the Es occurrence and its intensity vary strongly with season, their correlation with the S4 found on the ground are of great interest for scintillation alert and prediction. Here, we employ the RO S4 at $h_t$ = 30 km as a reference to infer the scintillations induced by E–region irregularities on the ground.

As shown in Figure 3, July from 2006–2014 is the month with a strong GNSS S4 from Es and its occurrence frequency peaks at the summer mid–latitudes. The diurnal and semidiurnal tidal modulations of Es occurrence were well documented in the GNSS observations [2–4]. For example, at 36° N latitude the semidiurnal Es variations show a tilted tidal structure with two maximums near $h_t$ = 100 km in 08:00–12:00 LST and 18:00–22:00 LST. At $h_t$ below the peak height, the S4 intensity decreases gradually with a long extension in the similar local time periods to the Es occurrence. It has

been speculated that the extended Es influence would contribute the scintillations as observed on the ground. In fact, the RO S4 statistics at $h_t$ = 30 km are quite consistent with the seasonal and diurnal variations from mid–latitude satellite–to–ground VHF/UHF (Very/Ultra High Frequency) scintillations [31–33]. A complete 12–month climatology of the S4 diurnal variation can be found in Figure S2 (supplementary). Generally speaking, the Es–induced S4 scintillations and their impacts on the ground–based receivers are stronger in the summer hemisphere with a significant diurnal variation.

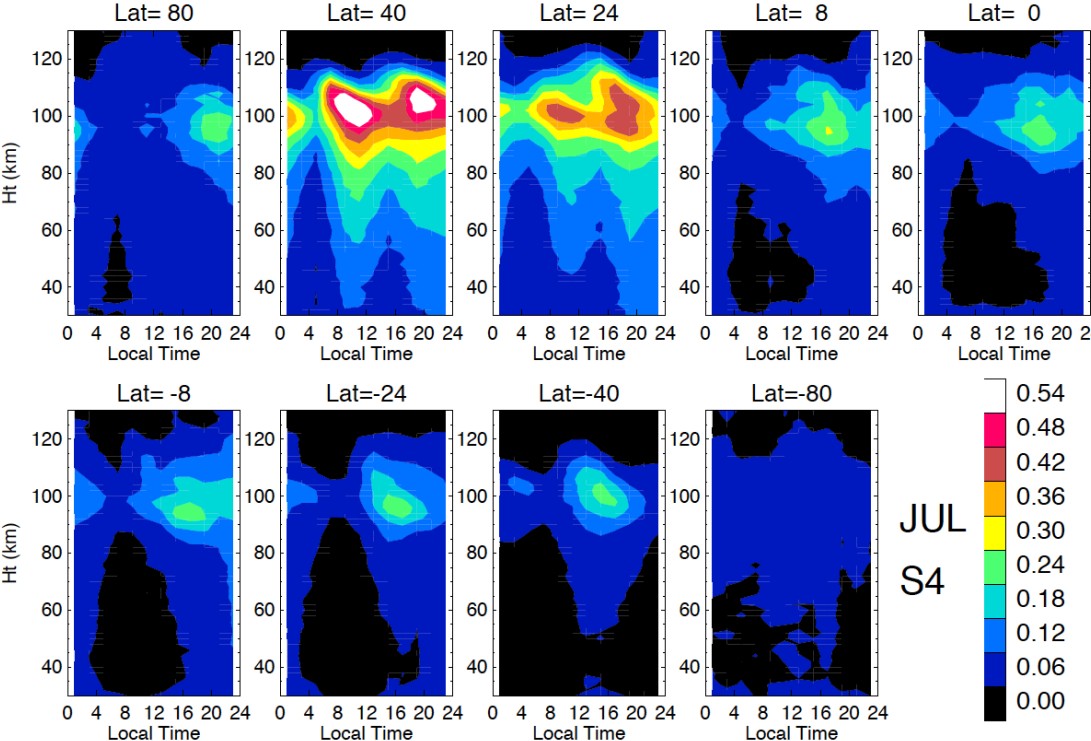

**Figure 3.** Diurnal variations of COSMIC–1 S4 from 2006–2014 for the month of July at selected latitudes.

Observations from the ground–based GNSS network revealed pulse–like Es disturbances in the mid–latitude TEC (Total Electron Content) enhancement [34], which appear as an elongate horizontal structure traveling like a front in the tropospheric weather system. These short–lived pulse–like Es structures also cause a burst of QP oscillations in GNSS signals as the Es enhancement travels across the ground–to–satellite link [32]. While atmospheric gravity waves are thought to play a key role in seeding the small–scale QP density fluctuations embedded in Es layers [27,35], it remains unclear what process leads to the solitary frontal–like Es lines as seen by the ground–based GNSS network [34].

The tropical and winter–hemispheric Es scintillations are dominated by a diurnal variation with the peak in the late afternoon [2,3]. Despite their weak amplitudes, these Es scintillations were found to occasionally affect the ground receivers [36]. Scintillation studies from the ground–based receivers near the magnetic equator showed that the daytime scintillations are often associated with the Es occurrence observed by COSMIC–1 [21]. These ground–based scintillation reports are consistent with the monthly S4 climatology revealed at $h_t$ = 30 km (Figure S2).

*3.2. Equatorial Plasma Bubbles (EPBs) and Equatorial Spread–F (ESF)*

Two major forms of ionospheric irregularities are equatorial plasma bubble (EPB) and equatorial spread–F (ESF). EPBs are a large–scale depletion of F–region electron densities initiated at the bottomside and extended up to the topside of the F–layer. ESF is an irregular backscatter signature on ionograms from the nighttime F–layer. The cause of ESF and EPBs can be explained in terms of Rayleigh–Taylor (R–T) instabilities [37] in the bottomside of the F–layer. Because these irregularities are associated

with a wide range of scales and structures, the induced scintillations have a large impact on both satellite–satellite [38] and ground–satellite [38] radio links.

Figure 4 shows such a wide spread of the scintillations from ESF and EPBs with an impact on the entire RO scan in the post–evening hours in March COSMIC–1 S4. At 20° N the S4 amplitude from ESF and EPBs remains approximately same above ($h_t$ = 120 km) and below ($h_t$ = 30 km) the Es layer, suggesting little impact form the RO path integration effect as discussed in Figure 1. In other words, these scintillations are not induced by simple layered or tilted structures. Rather, the near–uniform spread S4 amplitude suggests a broad spectrum of irregularity gradients in ESF and EPBs.

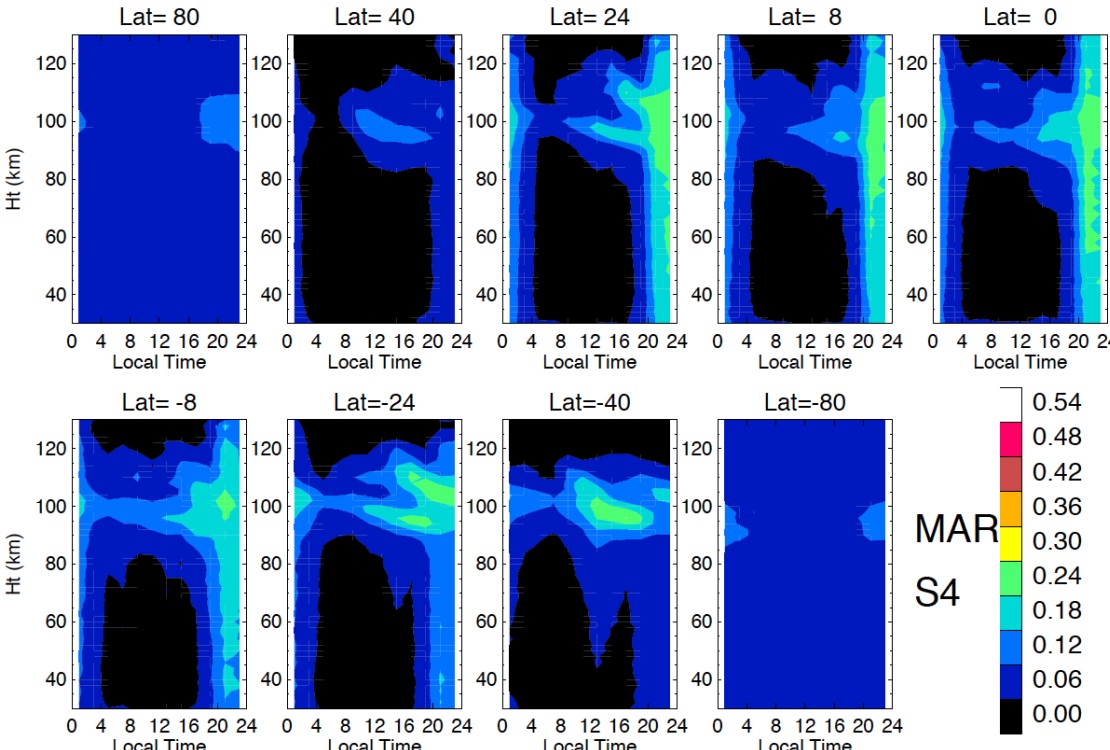

**Figure 4.** As in Figure 3, but for March from 2006–2014. The scintillations from equatorial spread–F (ESF) and equatorial plasma bubbles (EPBs) extend the RO slant path at $h_t$ = 30 km.

The scintillations from ESF and EPBs are characteristically different from those from Es layers and those with embedded QPs. The Es scintillation is relatively weak in March, most of which do not extend down to $h_t$ = 30 km. As in July, the Es scintillation amplitude decreases gradually with height from the Es peak altitude. The QP structures embedded in Es layers help the scintillation extension to a lower height, but they do not appear to have a wide structural spectrum to produce a uniform S4 power at all heights. However, the amount of S4 power reaching to a lower height (e.g., $h_t$ = 30 km) or to the ground will depend on the intensity of Es, which is a strong function of season and latitude.

Scintillations from ESF and EPBs have a different seasonal variation from Es as reported in previous studies [7,13,15,39–44]. As shown Figure 5, the RO S4 at $h_t$ = 30 km can be used to map the global distribution and seasonal variation of the nighttime (20:00–02:00) as observed from the in–situ sensors as well as from the ground–based receivers. Because of the extensive impacts seen in Figure 4 from ESF and EPBs, it is expected that this type of scintillations would extend further to ground–based receivers. In other words, the scintillations from a RO slant path (e.g., $h_t$ = 30 km) are perhaps more closely related to those observed from the ground. The consistent S4 distribution and seasonal variations between the RO slant path at $h_t$ = 30 km and the S4 climatology from other ESF/EPBs observations further support their connection.

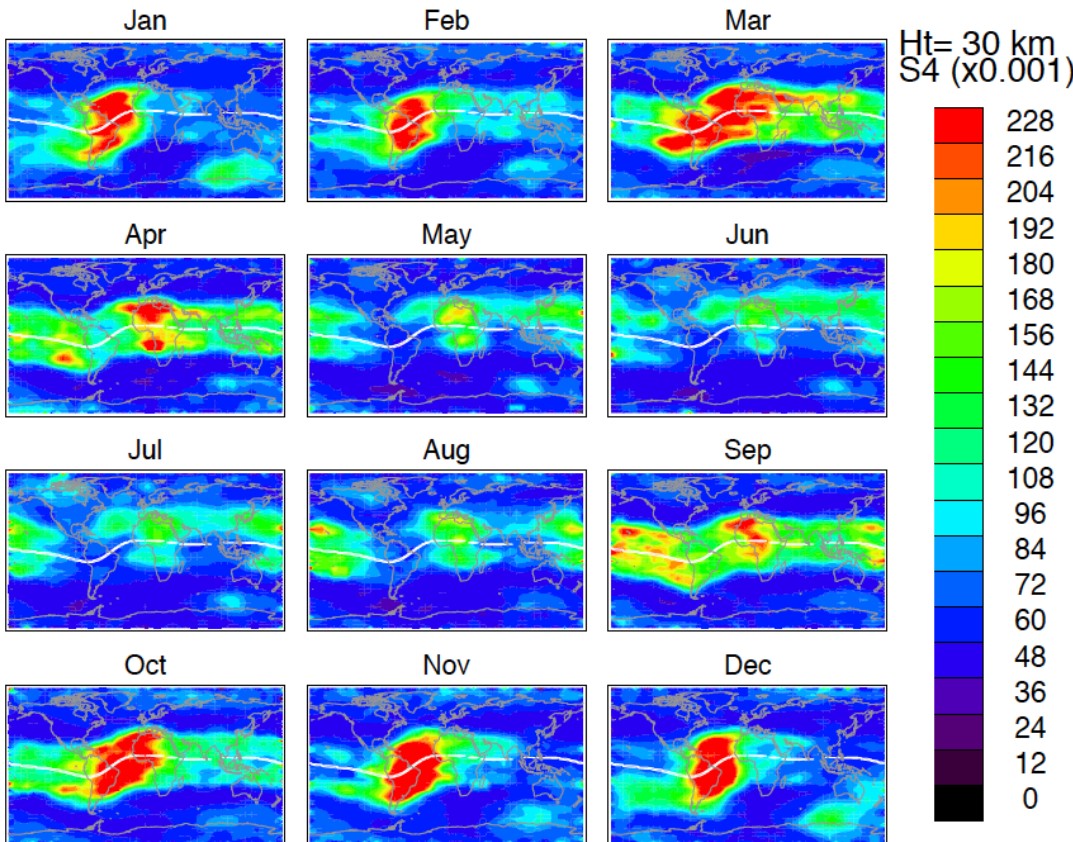

**Figure 5.** Monthly maps of COSMIC–1 S4 from 2006–2014 for 20:00–02:00 LST at $h_t$ = 30 km, showing dominant contributions from the nighttime scintillations induced by ESF and EPSs. The geomagnetic equator is denoted by the white line.

As a further evidence of the ESF/EPBs–induced scintillations, Figure 6 shows a seasonal variation of the RO S4 from ht = 30 km as a function of month and longitude in five selected local time bins. The patterns in Figure 6 match the seasonal variations of ESF/EPBs occurrence from the in–situ measurements at various satellite altitudes [29,45–47] as well as those on the ground [43]. The RO S4 variations appear to agree better with the vertical ion velocity (Vz) than the EPB occurrence frequency derived from plasma density anomalies in logarithmic scale [45], showing the seasonal asymmetry with a slightly higher mean in February–March than November–December. It was also found with the satellite in–situ measurements that the S4 amplitudes are correlated better with density–based EPB detection (i.e., ΔN) than with log(density)–based detection (i.e., ΔN/N0) [43,46]. The ΔN/N0–based detection appears to be more sensitive to the post–midnight EPB events than the N–based method [46,47]. In summary, the RO S4 results at ht = 30 km support the conclusion on the better correlation between S4 and the ΔN–based EPB detection.

### 3.3. Polar Scintillations

Different from the mid–latitude and equatorial irregularities, polar ionospheric scintillations are associated with diverse ionospheric/magnetospheric processes including polar cap patches, auroral blobs, auroral particle precipitation, and fronts of the ionospheric trough. Ground–based receivers in the polar region are more sensitive to phase ($\sigma_\phi$) scintillation than amplitude (S4) scintillation [48]. Observations show that approximate 80% of the cases are $\sigma_\phi$–only scintillations, 11% of S4–only scintillations, and ~9% with both [49]. In the northern hemisphere (NH), two geomagnetic latitude bands (65 ° N and 80 ° N) are associated with a high occurrence of S4 scintillations [50]. The 65 ° N band exhibits a significant seasonal variation with more scintillation activity in summer

and autumn, and a diurnal variation with the peak in midnight, whereas the high–latitude diurnal variations show a peak in post–midnight hours [49,50].

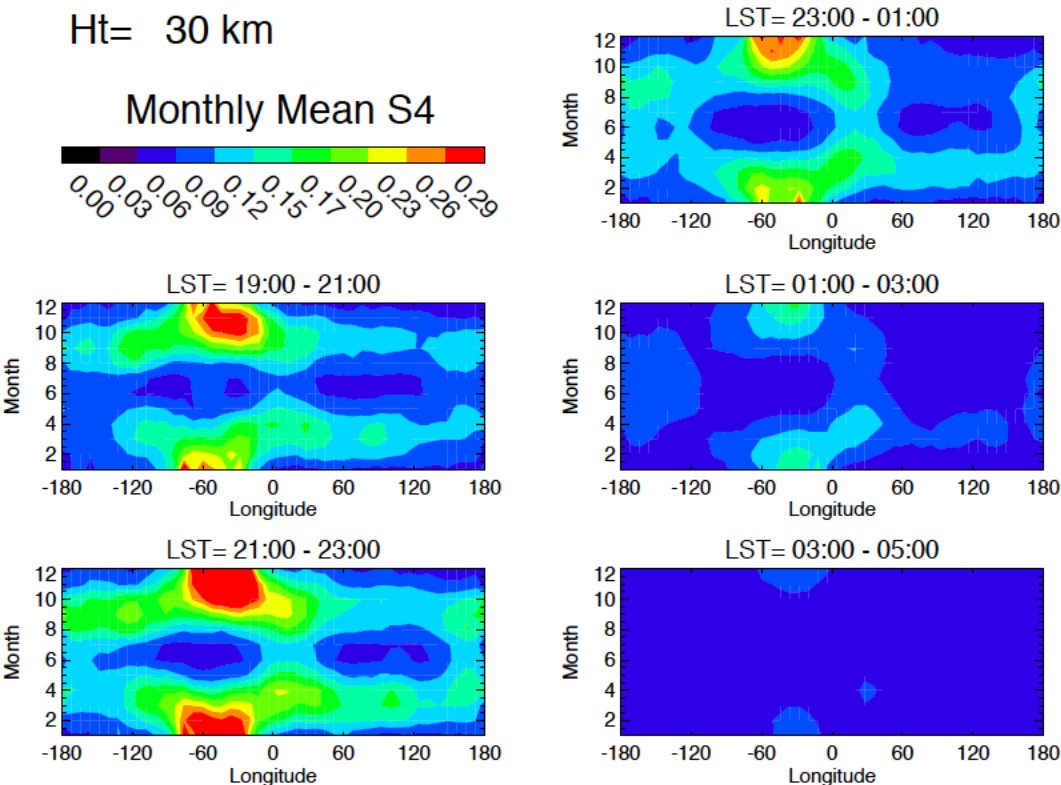

**Figure 6.** Seasonal variations of COSMIC–1 S4 from 2006–2014 at $h_t$ = 30 km for the geomagnetic latitude (Mlat) bin of 10° S–10° N.

As in the analysis for the equatorial S4 scintillations, we use the S4 at $h_t$ = 30 km as a proxy for the polar scintillations that would be observed on the ground. We find that the overall S4 scintillations in the polar regions are weak in all season. To help understanding what contribute the S4 at $h_t$ = 30 km, additional maps are made at $h_t$ = 100 km for the Es characteristics and at $h_t$ = 126 km for those contributions from a higher altitude. As shown in Figure 7 for the July S4, the Es has a significant impact on the scintillations at $h_t$ = 30 km, with most of the occurrence at magnetic dip angles between 20° and 60°. The NH Es $h_t$ = 100 km has a strong semidiurnal variation at mid–latitudes and a diurnal variation at high latitudes, as seen in Figure 3, which are likely to extend down to the ground level. In the SH, the Es is weak and does not extend much down to $h_t$ = 30 km. Rather, the scintillation map at $h_t$ = 30 km appear to have more influence from processes at a higher altitude with a pattern similar to that at $h_t$ = 126 km.

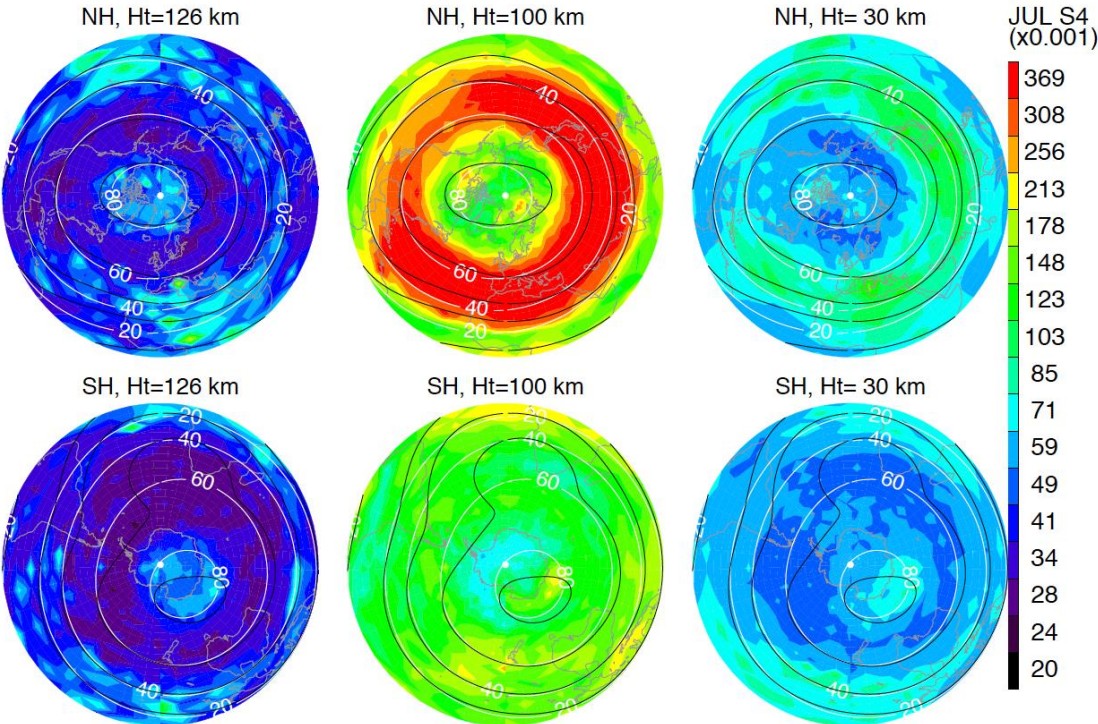

**Figure 7.** COSMIC–1 July S4 from all local times of 2006–2013 in the polar regions: northern hemisphere (NH) (top) and southern hemisphere (SH) (bottom). Black contours are the magnetic dip angles at 100 km (100km–dip angles) in the E–region ionosphere, whereas white contours ($R_E$–dip angles) are those at one Earth radius ($R_E = 6400$ km) in the lower magnetosphere.

At higher latitudes where the polar cap is located, the scintillations at $h_t = 126$ km a weak enhancement at the $R_E$–dip angles $> 80°$. The fact that these enhancements are more confined by the $R_E$–dip angles than by the 100km–dip angles suggests that they are likely originated from the magnetosphere rather than from the E–region ionosphere. In the NH these magnetosphere–originated enhancements at the $R_E$–dip angles $> 80°$ appear to extend down to $h_t = 30$ km. In the SH there is a signature of the S4 scintillation from the auroral oval at $h_t = 126$ km, and the scintillations at $h_t = 30$ km seem to have contributions from both the E–region and higher altitudes. The S4 enhancements associated with the polar cap are also seen in the climatology from other months (Figure S3 – supplementary), perhaps more evident in the months around the winter season (October–March).

### 3.4. Solar–Cycle Variations

Characterization of the long–term S4 variability requires a consistent satellite operation with uniform sampling and stable sensor performance. These conditions help to avoid sampling biases aliased into the long–term trend or variation in the measurements. Unfortunately, as shown in Figure 8, the COSMIC–1 constellation did not maintain such stability in sampling as the satellites/sensors in the constellation began to fail since 2014. On the other hand, the GPS–RO instruments on the MetOp series have been operating consistently since 2007 from a sun–synchronous orbit at 9:30 AM and 9:30PM equator crossing times. In particular, the MetOp–A operation has been very stable, providing ~680 ROs per day. The rising and setting ROs yield two slightly different local times on each node, which are separated by ~3h. MetOp–A started to drift slowly from the sun–synchronous orbit since 2017 (Figure 8), but it still stays roughly within the same equator–crossing local times. Hence, we use the MetOp–A data to study the solar–cycle variation of S4 scintillations.

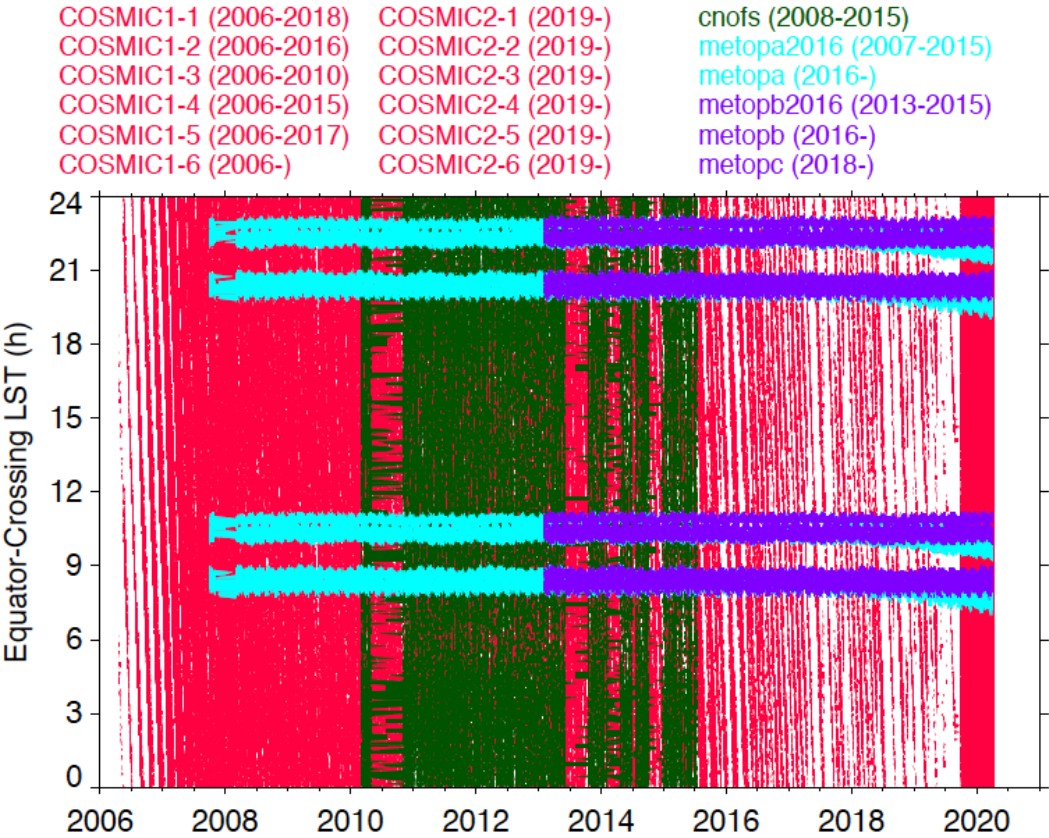

**Figure 8.** Equator–crossing local times from COSMIC–1, COSMIC–2, Communication/Navigation Outage Forecasting System (C/NOFS), and MetOp–A/B/C.

One of the MetOp local times (20:00–23:00) cover the frequent occurrence of scintillations induced by EPB and ESF as shown in Section 3.2. Therefore, by aggregating the MetOp data from the ascending and descending orbits, it provides the time series of S4 variations at the morning and post–evening hours. As revealed in Figure 9, the daytime S4 variations at $h_t$ = 40 km are dominated by the extended Es, showing the summer mid–latitude peaks. There is a significant solar–cycle variation in the S4 amplitudes from Es, with a lower S4 during the solar maximum. This anti–correlation between the solar cycle and Es–induced S4 was also reported in [2]. However, at high latitudes, the daytime S4 variation is positively correlated with the solar cycle, and the winter months have a larger S4 value in general.

The nighttime S4 in Figure 9 exhibits a strong semiannual variation at low latitudes, of which the amplitude is positively correlated with the solar cycle. The semiannual variation has two peaks near the equinoxes, which is consistent with the report from earlier studies using the 2003–2004 [51,52] and 2000–2006 data [9]. These studies found an asymmetry between the equinoctial peaks, showing the amplitude from March–April (spring) is higher than one from September–October (fall). An equinoctial asymmetry is also seen in the MetOp–A S4 variation. However, the relative importance of two equinoctial peaks depends on the phase of the solar cycle. During the rising phase of the solar cycle (2009–2013), the spring S4 is generally higher than the value from the previous fall. During the declining phase, however, the fall S4 is usually larger than the value in the following spring. Finally, at high latitudes the nighttime NH S4 is larger the SH S4, and both are positively correlated with the solar cycle.

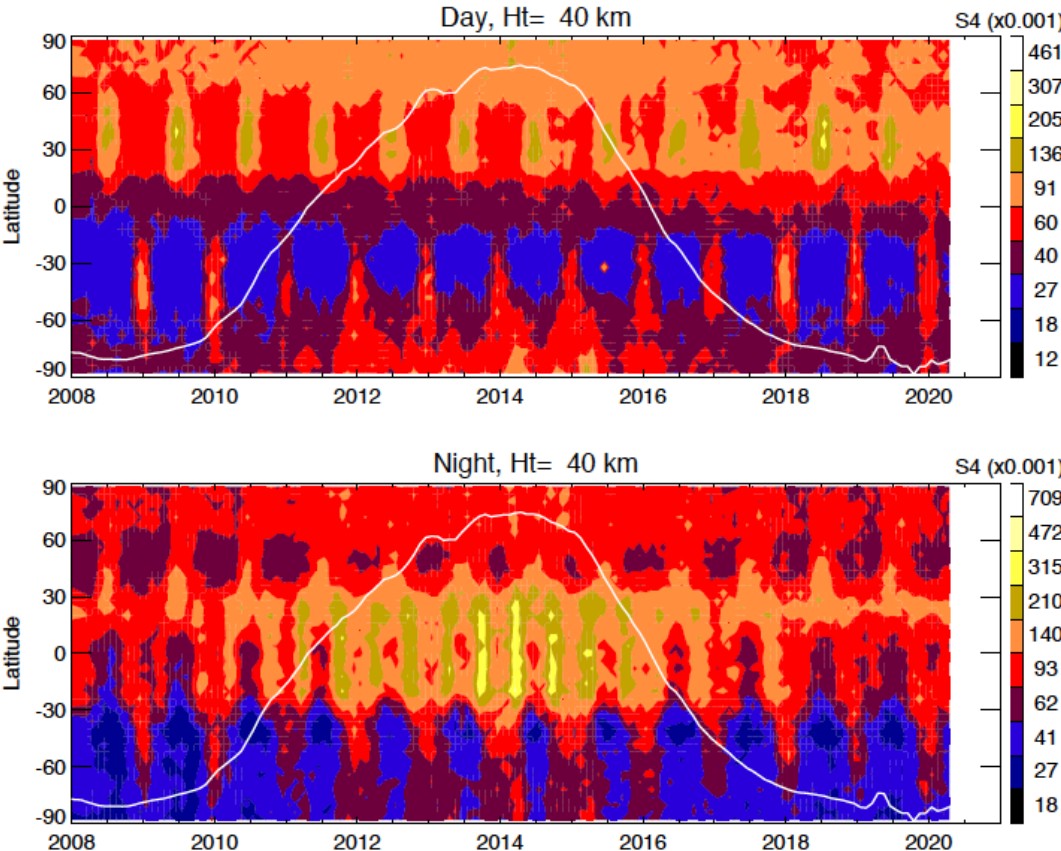

**Figure 9.** Time series of monthly MetOp–A S4 from 2008–2020 for day (top) and night (bottom) at $h_t = 40$ km. The white line is the scaled monthly F10.7 index for the solar cycle variation.

## 4. Discussions

### 4.1. Effects of Geo–Registration and Structured Disturbances

One of the limitations with the GNSS–RO technique is its poor horizontal resolution. As shown in Figure 1, the source of GNSS–RO scintillations can occur at either near or far side of the tangent point. Because of this ambiguity, we report all RO scintillations at the tangent point location. The approximation with the tangent point for geo–registration can induce an error of 2000 km in distance between the scintillation occurrence and the tangent point at $h_t = 30$ km (Figure 1). On the other hand, the geo–registration with tangent point may be more applicable than the actual location of disturbances for the scintillation forecast on the ground. In an attempt to establish transionospheric links at a 20° elevation angle, a ground receiver has the observing geometry similar to the GNSS–RO slant view at $h_t = 30$ km. This similarity implies that the upper F–region disturbances would make equivalent scintillation impacts on the ground–based and GPS–RO receivers.

To examine effects from the ambiguously geo–registered scintillations, we compare the monthly S4 maps at $h_t = 30$ km from COSMIC–1, COSMIC–2, and C/NOFS observations (Figure 10). Because these LEO satellites/constellations have very different orbital inclination angles (72°, 24°, 13°, respectively) and orbital altitudes (810, 715, and 400–600 km, respectively), the line–of–sight (LOS) direction of their RO links with respect to GNSS transmitters tends to differ from each other. The LOS from COSMIC–1 ROs tends to be more in the meridional direction, whereas the C/NOFS LOS are more in the zonal direction. The COSMIC–2 RO LOS falls between the two. The wider latitudinal distribution of ESF/EPB–induced scintillations from COSMIC–1 is likely resulted from the meridionally–dominated LOS in RO sampling. On the other hand, the narrow latitudinal band of S4 scintillations from C/NOFS reflects its zonally–biased LOS sampling. The S4 amplitudes are lower from COSMIC–2 because

its observations were mostly from a solar minimum period whereas both COSMIC–1 and C/NOFS covered a similar solar maximum period.

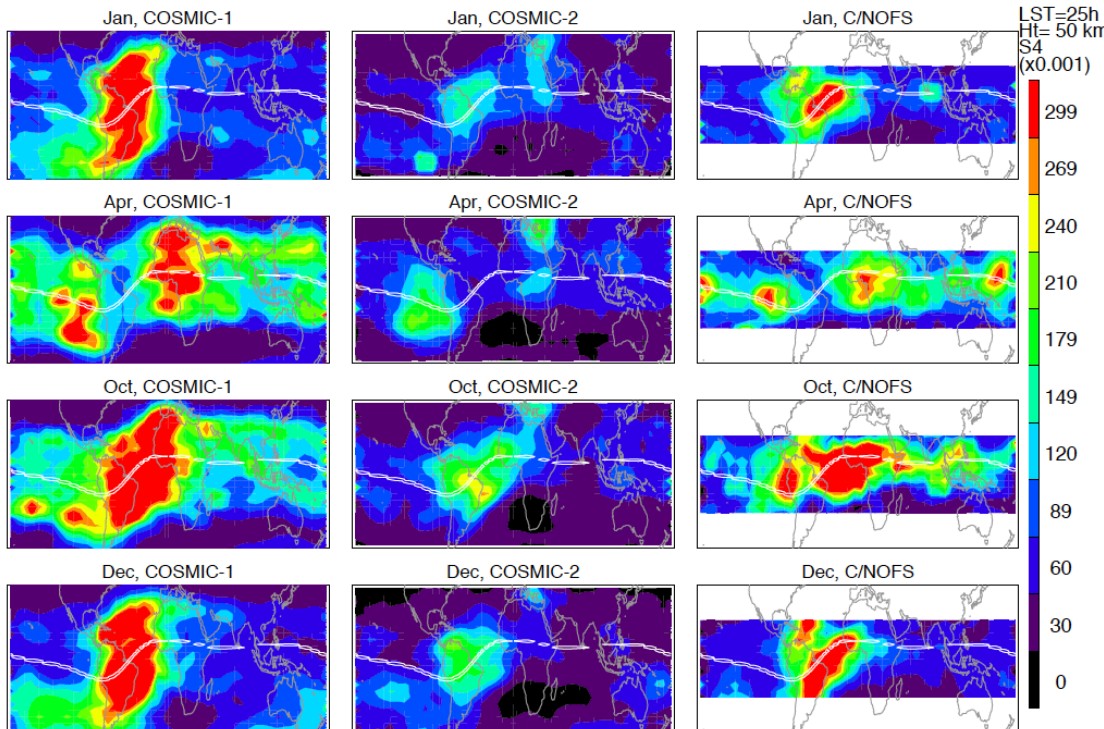

**Figure 10.** S4 maps from COSMIC–1, COSMIC–2, and C/NOFS for January, April, October, and December. The data period is 2006–2014 for COSMIC–1, 2019–2020 for COSMIC–2, and 2008–2015 for C/NOFS.

In addition to the geo–registration and solar–cycle effects, the S4 amplitude may also depend on the angle between the RO LOS and the magnetic field. The field–aligned plasma fluctuations, as seen in the radar echoes [27], can provide a highly–structured gradient embedded in an Es layer, which could induce scintillations in the situation where the RO LOS is parallel to the magnetic field. Compared to C/NOFS, COSMIC–1 satellites have a higher probability with a RO LOS aligned with the field line. As a result, COSMIC–1 is likely to observe more or larger S4 scintillations from the field–aligned disturbances.

### 4.2. GNSS–RO S4 and Phase ($\sigma_\phi$) Scintillations

In this study, we focused on the S4 scintillations and their impacts on GPS–RO and ground–based receivers. As discussed in the polar observation section, phase ($\sigma_\phi$) and amplitude (S4) scintillations may and may not occur simultaneously, and the dominance of these scintillations varies with location [48]. Here, the $\sigma_\phi$ scintillation is defined as the perturbation corrected by the dual–frequency approach, or the perturbation in the iono–free excess phase. Two cases are shown in Figure 11 for 1–s $\sigma_\phi$ and S4 scintillations: (top) elevated S4 at $h_t$ = 40–110 km but without a corresponding $\sigma_\phi$ enhancement in the height range; (bottom) simultaneous $\sigma_\phi$ and S4 scintillations at $h_t$ = 40–110 km. Errors from high–order ionospheric effects, multi–path propagation, and noisy phase measurements can lead to the phase variations that are uncorrectable with the dual–frequency (L1 and L2) method and cause scintillations in the iono–free excess phase measurement. To compute $\sigma_\phi$, in this study we first detrend the excess phase profile, as for the SNR profile, using the 1–s running mean; but the detrending procedure needs to be applied twice to the phase perturbation profile because the phase measurement

increases exponentially with decreasing $h_t$. The residuals between the running mean and the original measurement is called the excess phase perturbation.

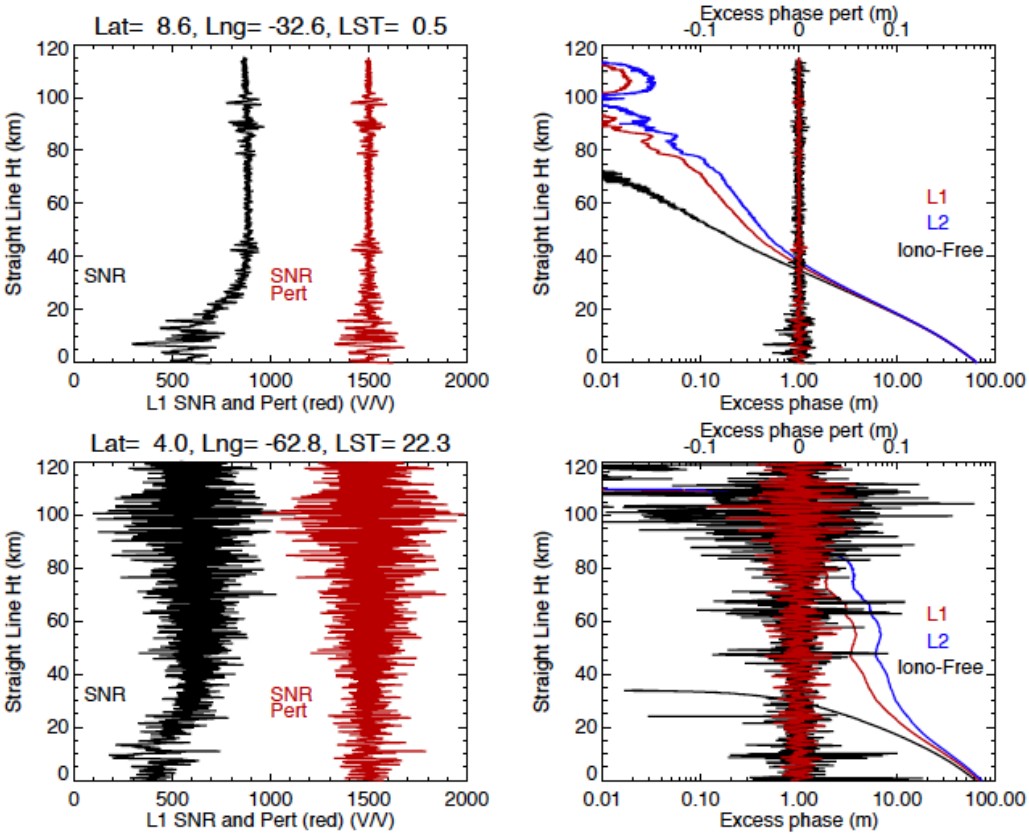

**Figure 11.** RO amplitude (S4) and phase ($\sigma_\phi$) scintillations that occur independently (top) and simultaneously (bottom). Signal–to–noise (SNR) perturbations, computed from the difference between the raw data and a 1–s running mean, are shifted by 1500 for the display. The phase perturbations are derived similarly, but by applying the differencing twice on the 1–s running mean to handle an exponentially varying profile.

Unlike S4, the $\sigma_\phi$ scintillations are perhaps more scale–dependent. Phase perturbations with a slow temporal variation are correctable with the dual–frequency (L1 and L2) approach. As seen in the top panels of Figure 11, the L1 and L2 phase profiles exhibit a similar fluctuation at 80–100 km, which are correctable by the differential method for the iono–free phase measurement. Uncorrectable phase fluctuations are generally considered as $\sigma_\phi$ scintillations. These large–scale or slow variations would be considered as $\sigma_\phi$ scintillations in the single–frequency phase measurement. However, because they are correctable with high–rate (50Hz) measurements, the iono–free phase data do not observe the elevated $\sigma_\phi$ scintillations as in the single–frequency measurements.

Scale–dependent scintillations need to be defined clearly in a scintillation analysis by specifying the spatiotemporal scale used to derive measurement perturbations. For example, in Figure 11, the $\sigma_\phi$ scintillation would show up with a significant amplitude if the running mean window were increased from 1–s to 4–s in the case of the top panel. This is typically the case where the source of perturbations has power from a narrow spectrum, such that scintillation power would increase when widening the filtering window. In the case of the bottom panel, where scintillations have a wide spectral power, changing the filter width would not impact the scintillation results significantly. A wide spectrum of plasma irregularities may also affect radio communications in the bands other than L–band. Elevated L–band S4 does not imply an increased S4 in other frequency bands. Hence, more studies are needed

to understand, quantify, and harmonize statistical connection between S4 and $\sigma_\phi$ scintillations from different spectral bands.

### 4.3. Implication for Scintillation Nowcast

GPS–RO S4 scintillations for slant–path views (e.g., $h_t$ = 30 km) provide useful observational constraints that are needed for the near–real–time (NRT), data–driven forecast of L–band communication outage [20]. Amplitude fluctuations can increase the rate of transmission bit error, causing data loss and cycle slips in the GPS signals. Severe S4 and $\sigma_\phi$ scintillations may result in loss of phase lock [30,53]. Interruption of services, in both satellite–satellite and satellite–ground links, ionospheric scintillation can significantly degrade the performance and the usability of space-based communication and navigation signals.

Because of the similar viewing/link geometry to ground–based receivers, the S4 scintillation derived from GPS–RO slant paths is expected to correlate better with those experienced on the ground, than the F–region scintillations derived from RO limb views. As discussed in Section 2, scintillations from the receiver on the ground with a link at 19° elevation angle is equivalent to the GPS–RO view at 400–km bubbles from $h_t$ = 30 km. Compared to the in–situ sensors at satellite orbital altitudes, which measure plasma disturbances from a Precise Orbit Determination (POD) receiver or magnetic/density sensors near the satellite position, GNSS–RO can sense the scintillations away from the orbital plane, and therefore provide a wider coverage. Because only a limited number of in–situ satellites can be used for NRT ionospheric bubble observations, the nowcast of global transionospheric scintillations from GPS–RO constellations becomes a viable and affordable solution, including implementation on commercial LEO constellations (e.g., Spire, Starlink, and Kuiper). As illustrated in Figure 12, a constellation of GNSS–RO receivers can establish a sensor web to provide a nowcasting system to alert satellite–to–satellite and satellite–to–ground communication outage.

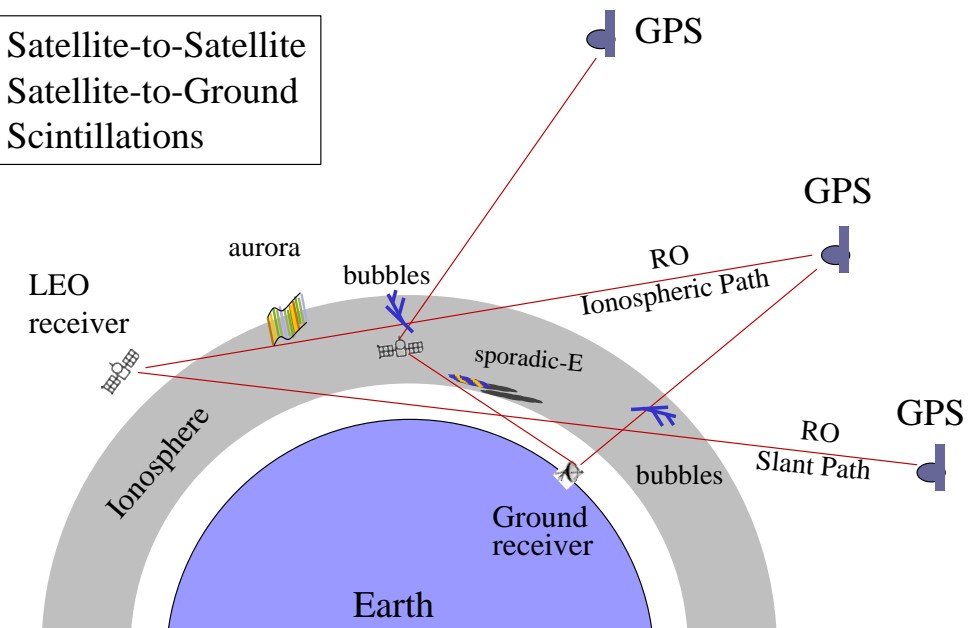

**Figure 12.** Schematic diagram of GPS receivers impacted by structured ionospheric disturbances. The scintillations detected by GPS–RO receivers can be relayed to a nowcast system to alert the ground–based receivers as well as LEO–based receivers.

## 5. Conclusions and Future Work

This study presents a climatology of 1–s L–band S4 scintillation from the 50–Hz GNSS–RO SNR data. The S4 at $h_t$ = 30 km, the lowest RO tangent height to measure ionosphere–induced

scintillations, appears to be a good proxy for the ground–based GNSS receivers in transionospheric links. The S4 characteristics identified in this study reveal important ionospheric sources of scintillation, as summarized in the following:

(1)  Es–induced S4 scintillations and their impacts on the ground–based receivers are stronger in the summer hemisphere with a significant diurnal variation;

(2)  GNSS–RO S4 scintillations from the slant path at $h_t = 30$ km show the geographical and seasonal variations similar to the observed climatology of ESF/EPBs, which are consistent with the scintillations found by ground–based receivers. The enhanced S4 varies significantly with local time and magnetic dip angle;

(3)  the polar S4 scintillations are weak in all seasons, as inferred from GNSS–RO S4 at $h_t = 30$ km;

(4)  a significant solar–cycle variation is found in the Es–induced daytime scintillations with a lower S4 value during the solar maximum. However, at high latitudes, the S4 variation is positively correlated with the solar cycle. The nighttime S4 exhibits a strong semiannual variation at low latitudes with a positive correlation with the solar cycle.

This study addresses primarily the 1–s L–band S4 scintillations induced by ionospheric plasma irregularities. The scintillations at other time scales and bands, as well as those in the phase measurements, warrant further investigations to better understand their connection to L–band S4 morphology.

In addition, direct comparisons between the ground–based and the LEO–based S4 measurements are feasible on a case–by–case basis, especially after the COSMIC–2 constellation was launched. Coincident scintillation observations may occur often with a ground station, LEO, and GNSS satellites in the same plane. As long as the ground station is in between LEO and GNSS with the LEO–GNSS slant path in parallel to the ground–GNSS link (Figure 1), it would yield a matched situation for comparing scintillations observed from the ground and from space. Further direct comparisons of S4 observations on a case–by–case basis will offer not only valuable insights on understanding scintillation processes but also a pathway for scintillation nowcast/alert on the ground with spaceborne measurements.

**Supplementary Materials:** The following are available online at http://www.mdpi.com/2072-4292/12/15/2373/s1, Figure S1: Monthly zonal mean S4 climatology from COSMIC–1 (2006–2014); Figure S2: Monthly climatology of S4 diurnal variations as a function of local time and $h_t$ from COSMIC–1 (2006–2014) at selected latitudes. Figure S3: Monthly polar maps of all–local–time mean S4 COSMIC–1 (2006–2013) at $h_t$ = 126, 100, and 30 km (Figure S1. As in Figure 2, but for all 12 months of the zonal mean S4 amplitude; Figure S2. As in Figure 3, but for all 12 months of the S4 diurnal variations; Figure S3. As in Figure 7, but for all 12 months of the polar S4 amplitude).

**Funding:** The work is by supported by NASA's Sun–Climate research at Goddard Space Flight Center (GSFC).

**Acknowledgments:** UCAR COSMIC Data Analysis and Archive Center (CDAAC) services for data processing and distribution.

**Conflicts of Interest:** The author declares no conflict of interest.

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
