# Peer review of "Ionospheric S4 Scintillations from GNSS Radio Occultation (RO) at Slant Path"

_remotesensing, doi:10.3390/rs12152373_

Round 1

Reviewer 1 Report

I have made the comments into the attached article.

Author Response

Your comments on the annotated manuscript have been incorporated in the revised paper.

Reviewer 2 Report

I have appreciated very much the study and recommend it for publication. 

I have just a few minor points to comment:

l. 55: What is the tangent height? I mean, there is the well know tangent point. Is it the same? If so, what is the definition of RO Top Ht (km) in Table 1? These terms needs more explanation for a proper understand. I could understand later in the paper but it was a bit confusing in the beginning. 

l. 62: what exactly are these excess phases?

l. 128: "statistics reported in other studies" [requires reference]

l. 347: I suppose you are talking about Figure 8 here and not 9, as mentioned.

More details are required to understand how you have computed the sigma-fi.

Figure 11: What is V/V?

Conclusions: In future I suggest performing a direct comparison between the LEO-based S4 measurements with ground-based S4 values. If this is hard to carry out due to geometry and coverage problems, I would recommend to indicate this.

Author Response

Please see the responses in bold font.

55: What is the tangent height? I mean, there is the well know tangent point. Is it the same? If so, what is the definition of RO Top Ht (km) in Table 1? These terms needs more explanation for a proper understand. I could understand later in the paper but it was a bit confusing in the beginning. 

A couple of sentences are added to clarify these variables:

“Here the RO tangent height is defined as the tangent point of straight-line height above the surface.”

“RO top ht is the approximately highest tangent height where the 50-Hz RO begins.”

  1. 62: what exactly are these excess phases?

The following sentences are added in the revision:

“The excess phase reported in the atmPhs and conPhs files is the additional phase delay/advance in RO technique, which is typically due to ionospheric/atmospheric effects, after the contributions from GPS transmitter and receiver satellite motions are removed. The correction still leaves an arbitrary constant in excess phase. As a result, the RO excess phase often references its profile to its top by setting the value at the top ht to zero.”

  1. 128: "statistics reported in other studies" [requires reference]

references have been added.

  1. 347: I suppose you are talking about Figure 8 here and not 9, as mentioned.

Correct. It has been changed to Fig.8.

More details are required to understand how you have computed the sigma-fi.

The following sentences are added in the end of the first paragraph in section 4.2:

“Here, the  scintillation is defined as strong perturbations not removed in the iono-free excess phase measurement. Errors from high-order ionospheric effects, multi-path propagation and noisy phase measurements can lead to the phase variations that are uncorrectable with the dual-frequency (L1 and L2) method and cause scintillations in the iono-free excess phase measurement. To compute , in this study we first detrend the excess phase profile, as for the SNR profile, using the 1-s running mean; but the detrending procedure needs to be applied twice to the phase perturbation profile because the phase measurement increases exponentially with decreasing ht. The residuals between the running mean and the original measurement is called the excess phase perturbation.”

Figure 11: What is V/V?

The following clarification is added at the section where SNR is introduced:

“The SNR from CDAAC is reported as a ratio in receiver voltage unit (V/V in 1 Hz) and the RO signal power is proportional to voltage amplitude squared.”

Conclusions: In future I suggest performing a direct comparison between the LEO-based S4 measurements with ground-based S4 values. If this is hard to carry out due to geometry and coverage problems, I would recommend to indicate this.

A very good suggestion. Added a new paragraph on this point:

“In addition, direct comparisons between the ground-based and the LEO-based S4 measurements are feasible on a case-by-case basis, especially after the COSMIC-2 constellation was launched. Coincident scintillation observations may occur often with a ground station, LEO and GNSS satellites in the same plane. As long as the ground station is in between LEO and GNSS with the LEO-GNSS slant path in parallel to the ground-GNSS link [Figure 1], it would yield a matched situation for comparing scintillations observed from the ground and from space. Further direct comparisons of S4 observations on a case-by-case basis will offer not only valuable insights on understanding scintillation processes but also a pathway for scintillation nowcast/alert on the ground with spaceborne measurements. “

Reviewer 3 Report

Abstract

  • 8/9 (prove statement): Ionospheric disturbances such as scintillation influence the propagation of radio signals and thus whether these signals can be received and processed by receivers. The availability of communication and navigation systems is different to the availability of communication and navigation services as well as to the usability of related signals.
  • 12-14 (prove statement): The L-band S4 from the RO measurements at ht = 30 km is related closely to ground-based scintillation observations due to the similar viewing geometry. Meaning of “related closely” and “due to similar viewing geometry” is too vague.

Introduction

  • 33/34 (explain statement in relation to the capability/characteristics of specific receiver): “The refractive scintillations may be correctable with a dual-frequency approach, if phase variations are slowly varying and the radio waves propagate along the same path….In addition, strong amplitude scintillations may result in loss of phase lock, producing uncorrectable phase measurement errors”

Data and Method

  • 116-118 (explain statement in relation to typical SNR values of GNSS, what means constant?, measuring unit is dB-Hz?): “Since the RO signal is nearly constant (???0) in the absence of scintillations, the normalization provides a fair characterization of scintillations between strong and weak RO signals. To minimize impacts of noisy signals on S4 statistics, the RO profiles with a low ???0 (???0 < 100) are excluded.”

Results

  • 232 and 247 (check figure reference)
  • Reference to Figure 8 is missed (347?)

Discussions

  • Figure 10/section 4.3: As mentioned the results of COSMIC 1 and 2 as well as C/NOFS are different due to different reasons e.g. by geospatial and solar cycle effects. However, it would be helpful to learn more about how to create a real comparability of results in order to clearly inform users of navigation and communication systems about their occurrence and influence.
  • 423 (ckeck wording): “Figure 11 shows that the 1-s sf and S4 scintillations sometimes occur simultaneously but sometimes not in the GPS-RO measurements.
  • 426-431 (please check the statement): “….These large-scale or slow variations would be considered as sf scintillations in the single frequency phase measurements, but would not for the iono-free phase measurements.” – here we see 2 options, a statement for the ionospheric effect (occurrence of scintillation) or a statement for the effects in measurements (scintillation/varying ionospheric delay can be corrected or not).
  • 440-442 (remark): It is well-known that the applied measuring methods (filter bandwidth of the receiver, calculation and averaging methods) codetermines the values of S4 and sf. Therefore approaches to harmonize/standardize such studies would be of common interest.
  • 455-457 (see 8/9)
  • Section 4.3 (450-474): It is undisputed that RO scintillation measurements can help to improve the near-time prediction of scintillation induced disturbances for ground-based GNSS receivers. However, this raises the question how much effort is needed (one LEO, one set of LEOs) to achieve a certain added value (probability that an existing scintillation will be detected). Not all ideas are feasible because of a possibly low cost-benefit ratio.
  •  

General remarks:

  • The figures should be near the text reference.
  • Check, if all abbreviations are explained e.g. LST
  • 502-504: please check the supplementary figures S1, S-3 and S-3

Author Response

Please see responses in italic font.

  • 8/9 (prove statement):Ionospheric disturbances such as scintillation influence the propagation of radio signals and thus whether these signals can be received and processed by receivers. The availability of communication and navigation systems is different to the availability of communication and navigation services as well as to the usability of related signals.

The sentence is changed to:

“Ionospheric scintillation can significantly degrade the performance and the usability of space‐based communication and navigation signals.”

  • 12-14 (prove statement): The L-band S4 from the RO measurements at ht = 30 km is related closely to ground-based scintillation observations due to the similar viewing geometry. Meaning of “related closely” and “due to similar viewing geometry” is too vague.

This is the topic sentence from a paragraph in the paper, to summarize the high-level message. Due to the word-count limit, we can’t expand the paragraph content here in the abstract.

For clarification the sentence is rewritten as:

“In this study the L-band S4 from the RO measurements at ht = 30 km is used to infer those from ground-based observations.”

Introduction

  • 33/34 (explain statement in relation to the capability/characteristics of specific receiver): “The refractive scintillations may be correctable with a dual-frequency approach, if phase variations are slowly varying and the radio waves propagate along the same path….In addition, strong amplitude scintillations may result in loss of phase lock, producing uncorrectable phase measurement errors”

This sentence perhaps introduced an unnecessary confusion on the nature and solutions to the phase scintillation problem. A number of solutions had been studied and implemented with single- and dual-frequency receivers, for which a comprehensive review can be found in Kintner et al. [2007] as in the reference.

The sentences are modified and reduced to the following:

“In the case where there is a rapid temporal variation in the plasma refractivity, fluctuations in the phase measurement are refractive, inducing phase (σ_ϕ) scintillation due to cycle slips or loss of phase lock. In reality, strong amplitude fluctuations may result in elevated phase measurement errors, for which amplitude and phase scintillations occur simultaneously.”

Data and Method

  • 116-118 (explain statement in relation to typical SNR values of GNSS, what means constant?, measuring unit is dB-Hz?): “Since the RO signal is nearly constant (???0) in the absence of scintillations, the normalization provides a fair characterization of scintillations between strong and weak RO signals. To minimize impacts of noisy signals on S4 statistics, the RO profiles with a low ???0 (???0 < 100) are excluded.”

A clarification sentence is added here:

“The SNR from CDAAC is reported as a ratio in receiver voltage unit (V/V in 1 Hz) and the RO signal power is proportional to voltage amplitude squared.”

So, 100 V/V-Hz = 20 dB-Hz

Results

  • 232 and 247 (check figure reference)

On L232 and L247, the reference to Fig.s2 is correct, but Fig.s2 was mislabeled.

The error has been corrected.

  • Reference to Figure 8 is missed (347?)

On L347 and L354 the reference should go to Fig.8, instead of Fig.9.

The error has been corrected.

Discussions

  • Figure 10/section 4.3: As mentioned the results of COSMIC 1 and 2 as well as C/NOFS are different due to different reasons e.g. by geospatial and solar cycle effects. However, it would be helpful to learn more about how to create a real comparability of results in order to clearly inform users of navigation and communication systems about their occurrence and influence.

This would be a premature task to sync the scintillation observations from different satellites, because the factors that affect the scintillation intensity are still not fully identified or characterized. For example, it’s possible that the scintillations might depend on the angle between radio wave propagation and geomagnetic field vector. With more GNSS receivers from LEO, however, it’s feasible to sort the scintillation measurements further in multi-dimensions, to better understand the inter-satellite differences.

  • 423 (ckeck wording):“Figure 11 shows that the 1-s sf and S4 scintillations sometimes occur simultaneously but sometimes not in the GPS-RO measurements.

The sentences are reworded as follows:

“Here, the σ_ϕ scintillation is defined as the perturbation corrected by the dual-frequency approach, or the perturbation in the iono-free excess phase. Two cases are shown in Figure 11 for 1-s σ_ϕ and S4 scintillations: (top) elevated S4 at ht = 40-110 km but without a corresponding σ_ϕ enhancement in the height range; (bottom) simultaneous σ_ϕ and S4 scintillations at ht = 40-110 km.”

  • 426-431 (please check the statement): “….These large-scale or slow variations would be considered as sfscintillations in the single frequency phase measurements, but would not for the iono-free phase measurements.” – here we see 2 options, a statement for the ionospheric effect (occurrence of scintillation) or a statement for the effects in measurements (scintillation/varying ionospheric delay can be corrected or not).

The description is further clarified as:

“Uncorrectable phase fluctuations are generally considered as σ_ϕ scintillations. These large-scale or slow variations would be considered as σ_ϕ scintillations in the single-frequency phase measurement. However, because they are correctable with high-rate (50Hz) measurements, the iono-free phase data do not observe the elevated σ_ϕ scintillations as in the single-frequency measurements.”

  • 440-442 (remark): It is well-known that the applied measuring methods (filter bandwidth of the receiver, calculation and averaging methods) codetermines the values of S4 and sTherefore approaches to harmonize/standardize such studies would be of common interest.

Appreciated!

  • 455-457 (see 8/9)

The sentence is corrected accordingly.

  • Section 4.3 (450-474): It is undisputed that RO scintillation measurements can help to improve the near-time prediction of scintillation induced disturbances for ground-based GNSS receivers. However, this raises the question how much effort is needed (one LEO, one set of LEOs) to achieve a certain added value (probability that an existing scintillation will be detected). Not all ideas are feasible because of a possibly low cost-benefit ratio.

It would require a LEO constellation, which is commercially available (e.g. Spire). In addition, it wouldn’t cost much to other businesses (e.g. Starlink) to send a few bytes of S4 down from its GPS receivers if a high temporal sampling is needed.

General remarks:

  • The figures should be near the text reference.

A rearrangement is made.

  • Check, if all abbreviations are explained e.g. LST
  • 502-504: please check the supplementary figures S1, S-3 and S-3

Errors in label and caption are corrected.